

# Topic adversarial neural network for cross-topic cyberbullying detection

Shufeng Xiong[1], Wenzhuo Liu[1], Bingkun Wang[2], Yinchao Che[1] and Lei Shi[1]

[1] College of Information and Management Science, Henan Agricultural University, Zhengzhou, China
[2] School of Information Engineering, Zhengzhou College of Finance and Economics, Zhengzhou, China

## ABSTRACT

With the proliferation of social media, cyberbullying has emerged as a pervasive threat, causing significant psychological harm to individuals and undermining social cohesion. Its linguistic expressions vary widely across topics, complicating automatic detection efforts. Most existing methods struggle to generalize across diverse online contexts due to their reliance on topic-specific features. To address this issue, we propose the Topic Adversarial Neural Network (TANN), a novel end-to-end framework for topic-invariant cyberbullying detection. TANN integrates a multi-level feature extractor with a topic discriminator and a cyberbullying detector. It leverages adversarial training to disentangle topic-related information while retaining universal linguistic cues relevant to harmful content. We construct a multi-topic dataset from major Chinese social media platforms, such as Weibo and Tieba, to evaluate the generalization performance of TANN in real-world scenarios. Experimental results demonstrate that TANN outperforms existing methods in cross-topic detection tasks, significantly improving robustness and accuracy. This work advances cross-topic cyberbullying detection by introducing a scalable solution that mitigates topic interference and enables reliable performance across dynamic online environments.

## INTRODUCTION

With the rapid development of social media, online platforms have reshaped not only how people access information and express opinions, but also become fertile ground for the emergence of harmful online behavior (*Oksanen et al., 2022*). Cyberbullying is one of the most prominent and pervasive issues. It refers to the intentional and repeated use of language in digital spaces to insult, demean, threaten, or humiliate others. Such behavior can cause serious psychological harm to victims, including anxiety and depression, and in severe cases, may even lead to self-harm or suicidal tendencies. Beyond individual consequences, cyberbullying also exacerbates social polarization and undermines public trust and social cohesion (*Anjum & Katarya, 2024*). The linguistic patterns of cyberbullying are highly diverse, ranging from overt insults to more implicit forms such as

Corresponding authors
Bingkun Wang, bkwang77@163.com
Lei Shi, shilei@henau.edu.cn

> **#Reading Books On The Subway#** I can understand playing on your phone while on the subway, but here you are reading a book, wearing headphones to listen to music, and even sporting a pair of glasses on your head. Are you trying to study or not? Or are you just trying to grab attention?
>
> ---
>
> 👤**User 1**: Are you stupid? How the hell does someone quietly reading a book affect your pathetic life?
>
> 👤**User 2**: None of your damn business! Are you that desperate to get insulted, you shit-eating mutt?
>
> 👤**User 3**: They're probably flipping through your family tree, figuring out where to start burying you.

**Figure 1 Typical examples of cyberbullying comments related to social justice on social media platforms.** These comments, translated from Chinese social media, illustrate topic-specific aggressive language and cyberbullying behaviors.

sarcasm, coded language, regional discrimination, and ideological bias. Moreover, its content often spans sensitive topics, including politics, religion, gender, ethnicity, and social identity. The linguistic diversity and contextual sensitivity of cyberbullying pose significant challenges for its automatic detection, especially within natural language processing frameworks (*Jahan & Oussalah, 2023*).

The complexity of cyberbullying is further exacerbated by topic variability. Controversial topics such as politics, religion, gender equality, and social justice often provoke emotionally charged and polarized discussions, prompting users to express aggressive opinions and hostility toward opposing views (*Akter et al., 2023*). For instance, users may attack individuals merely for reading a socially sensitive book in public or for being associated with certain ideologies or identities. In such cases, even neutral or unrelated behaviors may become targets of online aggression triggered by topic sensitivity. Figure 1 illustrates how cyberbullying content frequently intertwines with specific topics, further complicating detection by traditional methods. Furthermore, models trained on topic-specific datasets often fail to generalize across domains, with performance sharply declining when applied to new or unrelated content. This phenomenon, known as topic shift, underscores the need for robust models capable of filtering out topic-dependent features and capturing topic-invariant linguistic cues (*Kumar et al., 2024b*).

To address the challenge of cyberbullying detection, a variety of computational approaches have been explored in recent years. Early methods were primarily lexicon-based or relied on traditional machine learning algorithms (*Mohapatra et al., 2021*; *Chia et al., 2021*), while more recent advances have leveraged deep learning architectures such as convolutional neural networks (CNN), long short-term memory networks (LSTM), and Transformer-based models (*Obermaier & Schmuck, 2022*; *Zheng et al., 2024*). Although these models perform well within narrowly defined domains or specific datasets,

their performance often deteriorates significantly when applied to content from different topics or platforms. This is largely due to their heavy reliance on topic-specific linguistic patterns, which limits their generalization ability across diverse online contexts. To mitigate this issue, techniques such as domain adaptation and adversarial learning have been introduced to extract domain-invariant features and improve model transferability (*HassanPour Zonoozi & Seydi, 2023*). However, most existing methods do not explicitly address the challenge of topic-level variability, and there remains a lack of dedicated frameworks that can disentangle topic-specific features from universal indicators of cyberbullying. In particular, achieving fine-grained semantic understanding while effectively eliminating topic-induced biases remains a non-trivial challenge.

To address the challenges posed by topic variability in cyberbullying detection, this study introduces a novel cross-topic detection framework, Topic Adversarial Neural Network (TANN), which integrates deep feature extraction with adversarial learning to improve model robustness and generalizability. Unlike conventional models prone to overfitting on topic-specific cues, TANN is designed to isolate and retain topic-invariant linguistic features while suppressing topic-dependent noise. The architecture consists of three core components: a multi-level feature extractor, a topic discriminator, and a cyberbullying detector. The feature extractor combines the representational power of bidirectional encoder representations from Transformers (BERT), CNN, and bidirectional long short-term memory (Bi-LSTM), enabling it to capture rich semantic and syntactic information at multiple levels of abstraction. The extracted features are then passed through a topic discriminator trained in an adversarial manner, thereby forcing the shared encoder to "unlearn" topic-specific signals. This adversarial training process enables the model to produce topic-independent representations that are more robust to domain shift. Finally, the cyberbullying detector utilizes these purified representations to classify harmful content with higher accuracy across diverse topical domains. Through this architecture, TANN effectively bridges the gap between topic-specific expressiveness and cross-topic generalizability, offering a scalable solution for real-world cyberbullying detection.

To evaluate the effectiveness and practical applicability of the proposed TANN framework, we constructed a real-world dataset sourced from mainstream Chinese social media platforms, including Weibo and Tieba. These platforms span a wide range of discussion topics and user communities, offering a rich and diverse linguistic environment for studying cyberbullying in real-world contexts. The dataset was carefully curated to include multiple topical domains, enabling a comprehensive assessment of cross-topic generalization performance. Experimental results on this dataset demonstrate that TANN significantly mitigates topic interference by learning to focus on universal linguistic patterns associated with cyberbullying. This not only improves the accuracy of cyberbullying detection across diverse topics but also underscores the practical value of the model in real-world deployment scenarios, where content diversity and topic shift are inevitable challenges.

The main contributions of this article are as follows.

1. We define the novel task of topic-invariant cyberbullying detection, identifying the generalization gap caused by topic-specific features in existing models.
2. We propose TANN, a unified neural architecture that combines BERT, CNN, and Bi-LSTM modules with an adversarial topic discriminator to disentangle topic information and improve classification performance.
3. We construct a real-world multi-topic benchmark dataset using data collected from major Chinese social media platforms such as Weibo and Tieba, enabling robust evaluation of cross-topic transfer capabilities.
4. We perform adversarial weight visualization and feature attribution analysis, demonstrating the ability of TANN to suppress topic-specific biases and capture universal cyberbullying cues.

The subsequent sections of this article are organized as follows. "Related Work" reviews related work and highlights the key differences between our proposed approach and existing methods. "Model" presents a detailed description of the proposed TANN model. "Experiments" outlines the experimental setup and reports the results along with in-depth analysis. "Discussion" discusses the implications and insights derived from the findings. "Conclusions" concludes the article and outlines directions for future research. Finally, the Funding Statement is provided at the end of the article.

## RELATED WORK

Recent research in cyberbullying detection has evolved significantly, encompassing traditional machine learning, deep learning, and adversarial learning approaches (see Table 1). Although notable progress has been achieved, substantial challenges persist, particularly concerning generalizability across diverse topical contexts and the nuanced identification of cyberbullying expressions. This section critically reviews the strengths and limitations of these methodologies, clearly positioning our work within this landscape and highlighting the novelty and scientific contributions of the proposed TANN.

### Machine learning

Machine learning (ML) has emerged as a cornerstone in automating cyberbullying detection, offering scalable solutions across diverse linguistic and contextual landscapes. *Raj et al. (2021)* conducted a comparative analysis of hybrid frameworks, evaluating 11 classification methods spanning traditional ML algorithms and shallow neural networks. Their findings revealed that logistic regression coupled with TF-IDF features delivered superior performance among conventional methods. In multilingual and code-mixed environments, ML techniques have been tailored to address linguistic complexity. For instance, *Kumar et al. (2024a)* examined Hinglish (Hindi-English) texts and effectively trained classifiers such as SVM and Random Forests using lexical, syntactic, and sentiment features to handle the hybridity inherent in code-mixed content. Similarly, *Mahmud, Ptaszynski & Masui (2024)* systematically evaluated ML models for low-resource languages such as Bangla and Chittagonian, emphasizing the necessity of language-specific

**Table 1 This table provides a comprehensive overview of representative datasets and relevant research for various types of models (machine learning, deep learning, adversarial learning).** The datasets are categorized by their specific characteristics: General refers to general cyberbullying datasets, Hate refers to hate speech datasets, and Others refers to general text classification datasets. The references cited under each model category encompass recent key studies conducted in the respective research fields.

| Model | Dataset | References |
|---|---|---|
| Machine learning | General | *Raj et al. (2021)*, *Kumar et al. (2024a)* |
| | | *Mahmud, Ptaszynski & Masui, 2024*, *Chia et al. (2021)* |
| | Hate | *Mohapatra et al. (2021)*, *Saifullah et al. (2024)* |
| Deep learning | General | *Aliyeva & Yağanoğlu (2024)*, *Kumar & Sachdeva (2022)* |
| | | *Anbukkarasi & Varadhaganapathy (2023)*, *Caselli et al. (2020)* |
| | Hate | *Saleh, Alhothali & Moria, 2023*, *del Valle-Cano et al. (2023)* |
| | | *Aluru et al. (2020)*, *Plaza-del Arco et al. (2021)* |
| | Others | *Lyu et al. (2023)*, *Liu et al. (2023a)* |
| | | *Zou & Wang (2023)*, *Wang et al. (2022)* |
| Adversarial learning | General | *Haidar & Ezzeddine (2024)* |
| | | *Yi & Zubiaga (2022)*, *Sarwar & Murdock (2022)* |
| | Hate | *Bashar et al. (2021)*, *Nourollahi, Baradaran & Amirkhani, 2024* |
| | | *Bose et al. (2022)* |
| | Others | *Li, Jian & Xiong (2024)*, *Xu et al. (2022)* |
| | | *Liu et al. (2023b)*, *Han et al. (2021)* |

preprocessing to manage morphological and syntactic variations. Collectively, these studies underscore the importance of adapting ML models to diverse linguistic landscapes.

To further address the challenges posed by limited annotated data, *Saifullah et al. (2024)* proposed a semi-supervised framework that combines meta-vectorizers, including TF-IDF and Word2Vec, with self-training algorithms. Their approach demonstrated how semi-supervised ML can reduce dependence on costly manual annotations while maintaining robustness. Despite their successes, ML methods inherently depend on expert-crafted features, which are often insufficient for capturing complex, evolving linguistic phenomena characteristic of cyberbullying across diverse topics. Furthermore, these methods require extensive labeled datasets, limiting their scalability and adaptability to emerging contexts and shifting topical dynamics. These issues highlight the need for more flexible, context-aware methods. Deep learning architectures are well-suited to meet this need because they can automatically learn features and build hierarchical representations (*Jahan & Oussalah, 2023*).

## Deep learning

Recent advancements in deep learning have significantly enhanced cyberbullying detection, with the workflow typically divided into two critical stages: feature extraction and classification. The feature extraction phase focuses on learning hierarchical representations from raw text, user behavior, or multimodal data to capture semantic, structural, and contextual features that are discriminative. The classification stage then

builds end-to-end discriminative models based on these features and incorporates optimization strategies to address challenges such as data sparsity and linguistic complexity. This subsection systematically reviews related studies through the lens of feature representation learning and classification model design.

In feature representation learning, researchers have improved models' ability to capture bullying semantics through multi-feature fusion and domain adaptation strategies. For instance, *Aliyeva & Yağanoğlu (2024)* leveraged a multi-source feature fusion method that combined Term Frequency-Inverse Document Frequency (TF-IDF) and unigrams with social metadata to enrich text representation. The rise of Transformer architectures propelled the development of context-aware feature learning. BERT, proposed by *Devlin et al. (2018)*, employed masked language modeling to achieve bidirectional context encoding, while *Caselli et al. (2020)* further fine-tuned the model on hateful community data to obtain domain-specific representations. In cross-lingual settings, *Aluru et al. (2020)* validated the cross-language feature transfer capabilities of multilingual models (mBERT/ XLM-R), whereas *Plaza-del Arco et al. (2021)* showed that a language-specific model (BETO) outperformed others in Spanish-language tasks. In Chinese scenarios, *Wang et al. (2022)* addressed homonymy and polysemy issues by integrating glyph and pinyin features, and *Lyu et al. (2023)* proposed a bidirectional graph attention network to capture complex word-character relationships.

In classification decision-making, architectural innovations and learning paradigm improvements have been proposed to tackle practical challenges. For instance, hybrid neural network architectures such as the Bi-GAC model integrated gated recurrent unit (GRU), attention, and capsule networks to enable multi-level decision-making (*Kumar & Sachdeva, 2022*). *Saleh, Alhothali & Moria (2023)* further enhanced detection by integrating BERT with hate speech-specific word embeddings, allowing the model to better capture the lexical nuances of hateful content. In low-resource scenarios, *Liu et al. (2023a)* adopted a few-shot learning framework to expand supervisory signals *via* semantic clustering, while *Zou & Wang (2023)* employed pseudo-labeling to enlarge the training dataset for semi-supervised BERT. Multimodal fusion strategies have also been widely explored. For instance, *del Valle-Cano et al. (2023)* jointly analyzed textual features and user behavioral patterns to improve classification accuracy. Furthermore, domain adaptation methods have proven highly effective in toxicity detection. *Pavlopoulos et al. (2020)* utilized a context-aware mechanism to dynamically adjust classification thresholds, and *Anbukkarasi & Varadhaganapathy (2023)* proposed a specialized classifier for Tamil-English code-switched text. Collectively, these methods continue to enhance the robustness and generalization performance of cyberbullying detection systems in complex real-world scenarios.

Yet deep learning models trained on specific topical domains often exhibit significant performance degradation when confronted with topic shift. They often overfit to topic-specific vocabularies and fail to generalize to unrelated topics. The central challenge remains to develop deep architectures that can capture topic-invariant features while preserving the fine-grained semantic distinctions essential for accurate cyberbullying detection.

## Adversarial learning

Adversarial learning has become a pivotal approach for improving the robustness of cyberbullying detection models against adversarial attacks and domain shifts. Recent studies have explored diverse methodologies to address these challenges. *Liu et al. (2023b)* developed hierarchical neural network and gradient reversal (HNN-GRAT), a Robustly Optimized BERT Approach (RoBERTa)-based adversarial training framework that stabilizes text classification models by dynamically fusing original and reversed gradients through gradient reversal layers. This method demonstrates enhanced resistance to malicious perturbations, particularly in scenarios involving crafted adversarial examples. For cross-domain generalization, *Yi & Zubiaga (2022)* proposed platform-aware adversarial encoding to eliminate platform-specific noise in social media data, enabling robust cyberbullying detection across heterogeneous platforms like Twitter and Weibo. *Han et al. (2021)* further advanced few-shot adaptation by integrating meta-learning with adversarial domain adaptation, where episodic task simulations and meta-knowledge generators jointly learn domain-invariant features for detecting bullying patterns in low-resource target domains.

Building on these strategies, structural innovations have also been introduced. *Li, Jian & Xiong (2024)* combined graph attention networks (GAT) with adversarial noise injection, leveraging triple GAT layers to model character-word relationships while using TextCNN-generated adversarial samples to improve model generalization. For defensive lexical processing, *Xu et al. (2022)* designed WordRevert, a Chinese-specific method that neutralizes adversarial perturbations through keyword reversal, effectively preserving semantic coherence under attacks. Data augmentation strategies have also gained traction, *Haidar & Ezzeddine (2024)* employed GANs to synthesize cyberbullying samples, mitigating class imbalance issues in real-world datasets, while *Sarwar & Murdock (2022)* generated domain-adaptive hate speech data *via* sequence labeling and template-based augmentation for unsupervised domain adaptation.

Continuing this line of inquiry, domain adaptation remains a critical focus. *Bashar et al. (2021)* implemented progressive domain adaptation to transfer hate speech detection capabilities from general corpora to COVID-19-related anxiety texts, using unsupervised feature alignment. *Nourollahi, Baradaran & Amirkhani (2024)* enhanced cross-domain generalization through mixture-of-experts and adversarial training, achieving consistent performance improvements across multiple hate speech datasets. *Bose et al. (2022)* addressed domain-specific term overfitting by penalizing source-dependent keywords *via* domain classifier-guided attribution, forcing models to prioritize cross-domain discriminative features like derogatory syntactic patterns.

However, current adversarial methods primarily focus on platform adaptation and do not effectively address topic shift, which is a more subtle yet pervasive issue. Most existing adversarial frameworks lack mechanisms explicitly designed to mitigate topic-specific vocabulary bias, which significantly hinders performance across diverse topical scenarios.

## Summary and contributions

Despite recent advancements, the literature on cyberbullying detection reveals clear gaps: (1) inadequate handling of topic shift due to reliance on topic-specific features, (2) limited understanding of the nuanced interplay between topic-specific and universal linguistic patterns, and (3) lack of explicit mechanisms in current adversarial methods to disentangle topic biases while preserving critical fine-grained semantic information.

To address these critical challenges, we propose a novel approach termed TANN. TANN introduces the formalization of "topic-invariant cyberbullying detection", which represents a significant conceptual innovation aimed at explicitly minimizing topic-specific biases. Our proposed framework integrates a multi-level feature extractor that combines BERT, CNN and Bi-LSTM with an adversarially trained topic discriminator, thereby effectively disentangling topic-specific information and reinforcing universal linguistic patterns associated with cyberbullying. Extensive experiments on Chinese social media datasets demonstrate that TANN is significantly more robust and accurate in cross-topic scenarios than existing methodologies.

In summary, TANN fills critical research gaps by explicitly addressing the pervasive challenge of topic shift, enhancing generalization across topical contexts, and overcoming limitations inherent in existing adversarial and deep learning approaches.

## MODEL

This section introduces the TANN model, designed to enhance cross-topic cyberbullying detection. The model consists of three key components: a text feature extractor, a cyberbullying detector, and a topic discriminator. We first describe the role of each component, followed by an explanation of how they work together to enable transferable feature learning. Finally, we discuss the design of the objective functions used in the adversarial learning process.

### Model overview

The TANN model is designed to improve cross-topic cyberbullying detection by learning transferable, topic-invariant feature representations. As illustrated in Fig. 2, it consists of three core components. These components operate within an adversarial-cooperative framework to detect cyberbullying content while ensuring generalization across different topics.

The feature extractor generates rich feature representations from the input text by capturing both local and global information. It leverages Transformer blocks to model contextual relationships, CNN layers to detect local n-gram patterns, and Bi-LSTM layers to capture sequential dependencies. This multi-level design enables the model to effectively process various linguistic aspects of the input. The extracted features are then passed to both the cyberbullying detector and the topic discriminator. The cyberbullying detector classifies content as harmful or non-harmful, while the topic discriminator promotes topic-invariant representations through adversarial learning. Through joint training, the model simultaneously optimizes the cyberbullying detection objective and the topic

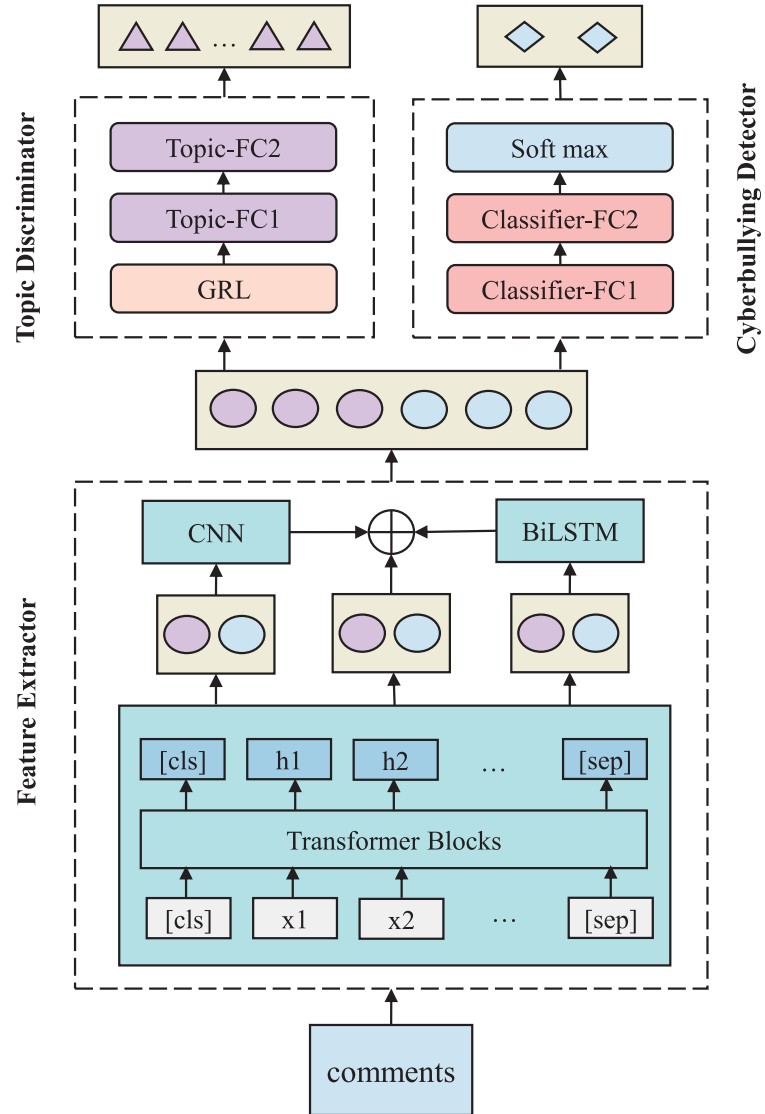

**Figure 2 TANN framework architecture, including the feature extractor, topic discriminator, and cyberbullying detector.** The feature extractor leverages BERT, CNN, and Bi-LSTM layers, while the topic discriminator uses a gradient reversal layer (GRL) to apply adversarial learning, reversing gradients during backpropagation to remove topic-specific features and enhance generalization.

invariance constraint, enabling accurate detection of harmful content across a broad range of topics.

## Feature extractor

The feature extractor processes each input, which includes a numeric label and text, by applying tokenization, padding, and preprocessing to prepare it for the model.

Each input text sequence is prepended with a special [CLS] token and appended with a [SEP] token, denoting the start and end of the sequence, respectively. These tokens guide

the contextual encoding performed by BERT. Formally, given a token sequence $x = \{x_1, x_2, \ldots, x_t\}$, the full input to BERT is $\{[\texttt{CLS}], x_1, x_2, \ldots, x_t, [\texttt{SEP}]\}$. BERT encodes this sequence and produces contextual embeddings $H_0 = \{h_1, h_2, \ldots, h_{t+2}\}$, where $h_i \in \mathbb{R}^d$ and $d$ is the hidden dimension. These embeddings are refined through $L$ Transformer layers, resulting in the final output $H_L = \{h_1^L, h_2^L, \ldots, h_{t+2}^L\}$. The hidden state corresponding to the $[\texttt{CLS}]$ token, $h_1^L$, is extracted as the sentence-level representation, as shown in Eq. (1).

$$F_{\text{bert}} = h_1^L. \tag{1}$$

The remaining BERT outputs $H_L' = \{h_2^L, h_3^L, \ldots, h_{t+1}^L\}$ corresponding to the input tokens (excluding special tokens) are used for downstream token-level feature extraction. These contextual embeddings are simultaneously fed into a CNN module and a Bi-LSTM to capture local patterns and sequential dependencies, respectively.

To extract local n-gram features, the CNN applies convolutional filters of various window sizes $k$. For a sliding window over token embeddings $H_L'$, the convolution operation is defined in Eq. (2).

$$C_i = \sigma(W_k \cdot H_{L,i:i+k-1}' + b_k). \tag{2}$$

Here, $W_k$ and $b_k$ are the convolution weights and biases for window size $k$, and $\sigma$ is a ReLU activation function. This operation produces a sequence of features given in Eq. (3).

$$C = [C_1, C_2, \ldots, C_{t-k+2}]. \tag{3}$$

Max-pooling is applied to each feature map to retain the most salient information. The resulting pooled vectors are concatenated into a single feature vector $F_C \in \mathbb{R}^{c \cdot n_k}$, where $c$ is the number of distinct window sizes and $n_k$ is the number of filters per size. This vector is then projected through a fully connected layer, as shown in Eq. (4).

$$F_{\text{cnn}} = \sigma(W_f \cdot F_C + b_f). \tag{4}$$

In parallel, the Bi-LSTM processes the token-level embeddings $H_L'$ to model sequential dependencies. For each token, the forward and backward hidden states are concatenated as shown in Eq. (5), forming the Bi-LSTM output $F_{\text{bilstm}} = \{h_1^b, h_2^b, \ldots, h_t^b\}$.

$$h_i^b = [\overrightarrow{h}_i; \overleftarrow{h}_i]. \tag{5}$$

A pooling operation is applied over the Bi-LSTM outputs to obtain a fixed-size vector representation $F_{\text{bilstm}} \in \mathbb{R}^{2h}$, where $h$ is the hidden size of each unidirectional LSTM.

Finally, the sentence-level feature from BERT, the CNN-extracted local features, and the Bi-LSTM-derived sequential features are concatenated to form a unified representation, as formulated in Eq. (6).

$$F_{\text{final}} = [F_{\text{bert}}, F_{\text{cnn}}, F_{\text{bilstm}}] = E_f(x; \theta_f). \tag{6}$$

The resulting feature vector $F_{\text{final}}$ combines global semantics, local patterns, and temporal dependencies. During training, a dropout layer with a rate of 0.1 is applied to $F_{\text{final}}$ to prevent overfitting and improve generalization.

## Topic discriminator

The topic discriminator is a neural network with two fully connected layers, each followed by an activation function. It classifies each comment into one of $K$ predefined topics using the feature representation $F_{\text{final}}$. We denote the discriminator as $D_t(F_{\text{final}}; \theta_t)$, where $\theta_t$ represents its parameters. The classification is guided by a cross-entropy loss function, where $Y_t$ is the set of topic labels.

Using the feature representation $F_{\text{final}}$, the discriminator computes the predicted probabilities for each topic $k$, denoted as $D_t(F_{\text{final}}; \theta_t)_k$. The topic discriminator is trained by minimizing the cross-entropy loss function $L_t(\theta_f, \theta_t)$, as shown in Eq. (7), where $\mathbf{1}[k = t]$ is an indicator function that is 1 if $k = t$, and 0 otherwise.

$$L_t(\theta_f, \theta_t) = -\mathbb{E}_{(x,t)\sim(X,Y_t)}\left[\sum_{k=1}^{K} \mathbf{1}[k = t] \log(D_t(F_{\text{final}}; \theta_t)_k)\right]. \tag{7}$$

The topic discriminator's parameters $\theta_t$ are optimized by minimizing $L_t(\theta_f, \theta_t)$, as shown in Eq. (8).

$$\hat{\theta}_t = \arg\min_{\theta_t} L_t(\theta_f, \theta_t). \tag{8}$$

The feature extractor is trained to learn topic-neutral representations through an adversarial mechanism. Specifically, a gradient reversal layer (GRL) is introduced between the feature extractor and the topic discriminator.

Formally, the GRL is defined as an operator $R_\eta(\cdot)$ that behaves as an identity function during forward propagation but reverses the gradient during backpropagation, as shown in Eq. (9).

$$F_{\text{final}}^{\text{GRL}} = R_\eta(F_{\text{final}}) = F_{\text{final}}. \tag{9}$$

However, during backpropagation, the GRL reverses the direction of the gradient and scales it by a factor $\eta$, as formally defined in Eq. (10).

$$\frac{\partial L_t}{\partial F_{\text{final}}} = -\eta \cdot \frac{\partial L_t}{\partial F_{\text{final}}^{\text{GRL}}}, \quad \eta > 0. \tag{10}$$

In practice, this reversal of gradients enforces adversarial learning, compelling the feature extractor to learn representations $F_{\text{final}}$ invariant to topic-specific information. The parameter $\eta$ controls the intensity of adversarial training. We empirically set $\eta = 0.1$, which yielded a good balance between representation generalization and task-specific performance, as further detailed in "Experimental Results Analysis".

Thus, the optimization objective for the feature extractor $\theta_f$ is effectively adversarial. It aims to maximize the topic discriminator's loss by minimizing the scaled negative topic loss as shown in Eq. (11).

$$\hat{\theta}_f = \arg\min_{\theta_f} - \eta \cdot L_t(\theta_f, \hat{\theta}_t). \tag{11}$$

This adversarial approach ensures the learned representations $F_{\text{final}}$ are robust and generalizable, reducing reliance on topic-specific features and thereby enhancing model performance across diverse topics (*Yao et al., 2022*).

## Cyberbullying detector

This subsection introduces the cyberbullying detector, which uses fully connected layers and a softmax function to classify comments as containing cyberbullying content or not. Building on the feature extractor, the detector takes the integrated feature representation $F_{\text{final}}$ as input. We denote the cyberbullying detector as $C_v(\cdot; \theta_v)$, where $\theta_v$ represents its parameters. For each comment $x$, the detector outputs a probability $P_\theta(x)$, indicating the likelihood that the comment contains cyberbullying content, as shown in Eq. (12), where $E_f(x; \theta_f)$ represents the features extracted by the feature extractor.

$$P_\theta(x) = C_v(E_f(x; \theta_f); \theta_v). \tag{12}$$

The primary goal of the cyberbullying detector is to accurately classify comments containing abusive content. We define a set of labels $Y_v$, where $y = 1$ indicates cyberbullying content and $y = 0$ indicates normal content. The cross-entropy loss function is used for classification, as defined in Eq. (13), to distinguish between the positive class (cyberbullying content) and the negative class (non-cyberbullying content).

$$L_v(\theta_f, \theta_v) = -\mathbb{E}_{(x,y)\sim(X,Y_v)}[y \log P_\theta(x) + (1-y)\log(1 - P_\theta(x))] \tag{13}$$

Optimization of the detector involves minimizing this loss to find the optimal parameters $\hat{\theta}_f$ and $\hat{\theta}_v$, as shown in Eq. (14).

$$(\hat{\theta}_f, \hat{\theta}_v) = \arg\min_{\theta_f, \theta_v} L_v(\theta_f, \theta_v). \tag{14}$$

A significant challenge in cyberbullying detection is the variability and evolving nature of abusive language. To generalize across different topics, the detector must capture core patterns of cyberbullying, independent of specific linguistic context. However, minimizing the classification loss on the training data may lead to overfitting to topic-specific patterns, such as certain keywords, which may not generalize well to unseen comments.

To overcome this, adversarial learning is introduced between the feature extractor and the detector. This setup encourages the feature extractor to learn more generalizable representations by reducing reliance on topic-specific details, thereby improving the robustness of the model. The detector learns features that capture universal characteristics of cyberbullying across diverse topics while minimizing the impact of context-dependent patterns.

## Adversarial-cooperative model integration

This subsection presents an adversarial-cooperative framework for cross-topic cyberbullying detection, integrating a feature extractor, topic discriminator, and cyberbullying classifier. These components are trained jointly with adversarial and cooperative objectives. The feature extractor $E_f(x; \theta_f)$ minimizes the cyberbullying

classification loss $L_v(\theta_f, \theta_v)$ in collaboration with the cyberbullying classifier $C_v(\cdot; \theta_v)$, while simultaneously maximizing the topic classification loss $L_t(\theta_f, \theta_t)$ with the topic discriminator $D_t(\cdot; \theta_t)$. This dual objective encourages the feature extractor to learn generalizable, topic-neutral features that enhance cyberbullying detection.

Formally, the cyberbullying classifier predicts the probability of a comment being bullying-related given the representation $F_{\text{final}}$, as defined in Eq. (15), where $W_v, b_v$ are the classifier weights and biases, and $\theta_v = \{W_v, b_v\}$.

$$C_v(F_{\text{final}}; \theta_v) = \text{softmax}(W_v F_{\text{final}} + b_v). \tag{15}$$

The overall adversarial-cooperative objective is defined in Eq. (16), where $\lambda \in [0, 1]$ is a hyperparameter controlling the trade-off between cyberbullying detection and topic classification. Based on parameter sensitivity analysis in "Experimental Results Analysis", we set $\lambda = 0.9$.

$$L_{\text{final}}(\theta_f, \theta_v, \theta_t) = \lambda L_v(\theta_f, \theta_v) + (1-\lambda) L_t(\theta_f, \theta_t). \tag{16}$$

The adversarial interaction between the feature extractor $F_{\text{final}}$ and the topic discriminator $D_t$ is implemented *via* a GRL, which behaves as an identity function during forward propagation but reverses the gradient direction during backpropagation. While $D_t$ minimizes the topic classification loss $L_t$ to identify topic-specific features, the feature extractor is trained to maximize $L_t$, thereby learning to obscure topic signals.

Parameter optimization is performed in two steps: the parameters of the feature extractor and cyberbullying classifier are updated jointly, as shown in Eq. (17), while the topic discriminator is updated independently as shown in Eq. (18).

$$\hat{\theta}_f, \hat{\theta}_v = \arg\min_{\theta_f, \theta_v} L_{\text{final}}(\theta_f, \theta_v, \hat{\theta}_t) \tag{17}$$

$$\hat{\theta}_t = \arg\min_{\theta_t} L_t(\theta_f, \theta_t). \tag{18}$$

This training strategy encourages the feature extractor to eliminate topic-dependent variations such as vocabulary biases in political *vs.* entertainment contexts, while preserving discriminative bullying indicators such as aggressive or sarcastic expressions. As a result, the model avoids overfitting to topic-specific noise, which is essential for detecting implicit bullying across different domains.

The adversarial objective promotes topic-invariant representations by aligning feature distributions in latent space across different topics. This alignment ensures that bullying and non-bullying instances form separable clusters regardless of their topic origin. Consequently, the cyberbullying classifier $C_v$ learns decision boundaries based on universal bullying traits rather than topic-specific cues. The model suppresses keywords uniquely associated with certain domains (*e.g.*, entertainment slang), while amplifying cross-topic bullying signals such as personal insults and coded language. Through end-to-end training, this framework improves both the accuracy and generalizability of cyberbullying detection across diverse topic domains.

# EXPERIMENTS

## Dataset

The dataset for this study was collected from two major Chinese social media platforms, Weibo (https://www.weibo.com) and Tieba (https://tieba.baidu.com), and comprises 24,520 user comments covering topics such as politics, entertainment, and social issues. Each comment is annotated with two labels: a theme label, categorizing it into one of eleven topics (0–10) to capture language variation across themes, and a cyberbullying label, a binary indicator classifying the comment as non-cyberbullying (0) or cyberbullying (1).

The theme labels were assigned based on specific keywords that categorize each comment into distinct topical domains. These topics include, but are not limited to, social issues, entertainment news, and personal viewpoints. The keywords associated with each topic were carefully selected to capture the core semantic content of the discussions. For instance, comments related to social issues often contain terms such as "poverty," "education," "healthcare," and "inequality," whereas those related to entertainment may include keywords like "celebrity," "movie," or "music." The keyword sets were constructed through an iterative process that combined domain expertise with linguistic analysis of the collected data.

For the cyberbullying annotation, the identification and labeling process followed a clear and systematic approach to ensure consistency. Instances of cyberbullying were identified based on common indicators of harmful online behavior, including offensive language, personal attacks, and harassment. We followed widely accepted definitions and criteria for cyberbullying, adapting them to the context of social media in China. A set of examples and guidelines was provided to annotators to help them classify comments as cyberbullying or non-cyberbullying. These guidelines were designed to ensure that borderline or ambiguous cases were consistently labeled by all annotators. The detailed annotation guidelines, including specific categories such as insults, threats, discrimination, sexual harassment, defamation, and satire, are publicly available along with the code for TANN.

The annotators involved in the labeling process were native Chinese speakers with backgrounds in linguistics, social media analysis, and psychology. Their expertise enabled them to accurately capture linguistic nuances on social media and reliably identify instances of cyberbullying. A total of five annotators participated in the labeling process. To assess the reliability of the labels, we calculated inter-annotator agreement using Fleiss' kappa coefficient, which yielded a value of 0.85, indicating a high level of consistency among the annotators. Despite the overall high consistency, we acknowledge that certain instances involving sarcasm, humor, and ambiguous language presented annotation challenges, which were addressed through regular discussions and refinements of the guidelines. In addition to the inter-annotator agreement, we employed quality control measures such as random sampling and having a second group of annotators re-label a subset of the data. This process helped identify discrepancies or inconsistencies in labeling and allowed us to correct labeling errors before finalizing the dataset for training.

**Table 2  Label distribution and data split in the cyberbullying dataset across training, validation, and test sets.** Label 0 indicates non-bullying comments, while label 1 indicates bullying comments.

| Label | Train | Test | Dev | Total |
|---|---|---|---|---|
| 0 | 14,955 | 1,908 | 1,862 | 18,725 |
| 1 | 4,661 | 544 | 590 | 5,795 |
| Total | 19,616 | 2,452 | 2,452 | 24,520 |

Before using the dataset for training, several preprocessing steps were applied to ensure data quality. The raw text was cleaned by removing extraneous elements such as special characters, hyperlinks, and excessive whitespace. The Chinese text was tokenized using a standard Chinese word segmentation tool, ensuring each comment was split into meaningful words or phrases. A list of common Chinese stopwords was applied to remove words that do not significantly contribute to the meaning of the text. To address potential class imbalance, under-sampling was applied to the non-cyberbullying class to balance the distribution between cyberbullying and non-cyberbullying labels.

The dataset was randomly shuffled and split into training, validation, and test sets with a ratio of 8:1:1, resulting in 19,616 samples for training, 2,452 for validation, and 2,452 for testing. This split ensures that the model is exposed to a wide variety of samples during training, thereby enhancing its ability to generalize. The distribution of cyberbullying labels across the training, validation, and test sets is shown in Table 2. This dataset forms the basis for training the proposed TANN.

Regarding dataset quality evaluation, in addition to traditional label verification, we also draw on evaluation methods proposed in existing literature. For instance, *Liu et al. (2022)* presented a framework for evaluating dataset quality, focusing on issues such as dataset representativeness, consistency, completeness, and noise. Their study highlighted errors encountered during the creation process of NIDS datasets and proposed methods for reverse-engineering and correcting the labeling logic. We have adopted similar quality evaluation methods to ensure that the dataset used in this study meets the required standards. The code for the TANN and the dataset used in this study are publicly available on GitHub (https://github.com/mugumulwz/Liu).

## Experimental setup

The experiments were conducted on a machine equipped with an NVIDIA RTX 4060 GPU, which provided the necessary computational power for training the TANN model. Training the TANN model required between 3 and 10 min per epoch, with the total duration varying from 6 min to 1 h depending on the parameter settings. The model was implemented using the PyTorch framework, with a final batch size of 32. The computational cost for each experiment remained between 0.1 and 1 GPU hour.

We employed a two-phase hyperparameter tuning strategy. In the initial phase, a grid search was conducted across learning rates from 1e−5 to 5e−5, epoch counts from 1 to 9, and dropout rates from 0.1 to 0.4. In the second phase, the batch size and padding length were optimized through iterative adjustment and evaluation on the validation set. Table 3

**Table 3 Overview of hyperparameter configurations evaluated for the TANN Model.** Bolded values indicate the selected settings used in the experiments.

| Hyperparameters | Values |
| --- | --- |
| Epochs | 1, 2, 3, 4, **5**, 6, 7, 8, 9 |
| Learning rate | 1e−5, **2e−5**, 3e−5, 4e−5, 5e−5 |
| Batch size | 4, 8, 16, **32** |
| Dropout rate | **0.1**, 0.2, 0.3, 0.4 |
| Padding length | 16, 32, **64** |

highlights the optimal configuration with bold values: five training epochs, a 2e−5 learning rate, a batch size of 32, a 0.1 dropout rate, and a padding length of 64. These settings were applied consistently across all baseline models, the TANN model, and ablation studies, ensuring fair comparisons. The total training time for each experiment was 15 min. This optimization process effectively balanced computational efficiency with model performance, achieving optimal detection results while preventing overfitting.

## Baseline models

To evaluate the effectiveness of the proposed model, we compare it against a diverse set of baseline models spanning traditional, neural, and transformer-based architectures. These baselines were carefully selected to represent a wide spectrum of text classification techniques, ranging from lightweight models with shallow structures to state-of-the-art deep learning approaches. Specifically, the baselines include traditional word embedding methods (FastText), sequence- and convolution-based neural models (TextRNN, TextCNN), pre-trained transformer encoders (BERT), and several hybrid architectures that combine BERT with CNN, RNN, adversarial training, or instruction-tuned language modeling (BERT-CNN, BERT-RNN, BERT-FGM, FLAN-T5). This comprehensive set of models provides a robust benchmark for assessing the performance, generalization, and robustness of our proposed TANN framework in the context of cross-topic cyberbullying detection. The baseline models, presented in order of increasing architectural complexity, are as follows.

- **FastText:** FastText is an open-source library developed by Facebook AI Research that focuses on efficient learning of word representations and text classification. By incorporating subword-level information through character n-grams, FastText captures finer-grained semantic and morphological features and effectively handles out-of-vocabulary words. The lightweight structure of the model and rapid training process enable it to scale to large datasets while delivering high-performance results. Consequently, FastText has become a popular choice for various natural language processing tasks, including text classification, language modeling, and word similarity (*Umer et al., 2023*).
- **TextRNN:** TextRNN is a neural network architecture designed to handle sequential text data by leveraging recurrent layers, such as LSTM or GRU, to capture and retain contextual information across tokens. By processing each token in a text sequence while

taking into account previously encountered tokens, TextRNN can effectively learn meaningful patterns for a wide range of natural language processing tasks, including text classification, sentiment analysis, and language modeling. The sequential nature of this architecture allows it to handle varying sequence lengths and capture long-term dependencies, making it a robust choice for many text-oriented applications (*Liu et al., 2023c*).

- **TextCNN:** TextCNN is a convolution-based neural network architecture designed for text classification and other natural language processing tasks. It applies multiple filters of different kernel sizes to word embeddings, capturing diverse local features in text sequences. These features are then aggregated, often through max-pooling, into a fixed-length representation that is passed to a fully connected layer for classification. Due to its simplicity, efficiency, and robust performance on tasks such as sentiment analysis, spam detection, and topic categorization, TextCNN is frequently used as a strong baseline in NLP research (*Jiang et al., 2022*).

- **BERT:** BERT is a transformer-based language model that learns deeply contextual word representations by capturing both left and right contexts in all layers. Through the use of masked language modeling and next sentence prediction as pretraining objectives, BERT can be fine-tuned for various downstream tasks, such as question answering, sentiment analysis, and named entity recognition. Its bidirectional design provides a richer understanding of linguistic nuances, enabling it to achieve state-of-the-art performance on many benchmarks in natural language processing research (*Devlin et al., 2018*).

- **BERT-CNN:** BERT-CNN is a hybrid model that combines the contextual embeddings from BERT with a CNN to enhance text classification and other natural language processing tasks. By leveraging the ability of BERT to capture deep semantic relationships between words, the CNN component can effectively learn local, position-invariant features within the text. This synergy enables the model to handle long-range dependencies and nuanced linguistic structures, making BERT-CNN highly effective for tasks such as sentiment analysis, question answering, and entity recognition (*Wan & Li, 2022*).

- **BERT-RNN:** BERT-RNN is a hybrid architecture that integrates the contextual embeddings of BERT with an RNN, such as an LSTM or GRU. This combination allows the model to exploit the ability of BERT to capture rich contextual information at the token level, while the RNN component effectively handles sequential dependencies in the text. As a result, BERT-RNN excels in tasks like sentiment analysis, text classification, and sequence tagging, where both contextual understanding and temporal order are critical (*Tan et al., 2022*).

- **BERT-FGM:** BERT-FGM is a variant of the BERT model that leverages adversarial learning by incorporating the fast gradient method (FGM) into its training process. Specifically, FGM introduces small, calculated perturbations to the input embeddings during training, which simulates adversarial attacks. This forces the model to learn more robust and generalized features, effectively enhancing its resistance to adversarial examples and improving overall performance on NLP tasks (*Naseem, 2024*).

- **FLAN-T5 (Large):** The FLAN-T5 (Large) model is an advanced large language model that extends the T5 architecture through instruction-based fine-tuning, enabling it to achieve robust zero-shot performance across a wide range of tasks. This model uses modern LLM technology to handle complex tasks such as text classification, often with little or no additional task-specific training. Its innovative approach of converting every NLP problem into a text-to-text format not only simplifies the training process but also significantly enhances its ability to generalize to unseen tasks, making it a powerful tool in scenarios where labeled data is scarce (*Sheikhaei et al., 2024*).

## Evaluation metrics

Macro-Accuracy and Macro-F1 were selected as the primary evaluation metrics to comprehensively assess the performance of the model in detecting cyberbullying in social media comments, particularly considering the class imbalance in the dataset.

Macro-accuracy measures the average proportion of correctly classified samples across all classes, providing an overall indication of balanced classification performance. It is defined as shown in Eq. (19), where $t$ denotes the total number of classes, and $TP_i$, $FP_i$, and $FN_i$ represent the true positives, false positives, and false negatives for the $i$-th class, respectively.

$$\text{Macro}-\text{Accuracy} = \frac{1}{t}\sum_{i=1}^{t}\frac{TP_i}{TP_i + FP_i + FN_i}. \quad (19)$$

Macro-F1 is the harmonic mean of macro-precision and macro-recall, offering a balanced evaluation metric that equally weighs performance across all classes, regardless of sample distribution. This metric is particularly useful for imbalanced datasets, as it evaluates each class independently before averaging. Macro-F1 is calculated as shown in Eq (20), where macro-precision ($P_{\text{macro}}$) and macro-recall ($R_{\text{macro}}$) are defined in Eqs. (21) and (22).

$$\text{Macro}-\text{F1} = \frac{2 \times P_{\text{macro}} \times R_{\text{macro}}}{P_{\text{macro}} + R_{\text{macro}}} \quad (20)$$

$$P_{\text{macro}} = \frac{1}{t}\sum_{i=1}^{t}\frac{TP_i}{TP_i + FP_i} \quad (21)$$

$$R_{\text{macro}} = \frac{1}{t}\sum_{i=1}^{t}\frac{TP_i}{TP_i + FN_i}. \quad (22)$$

Macro-F1 is particularly relevant to this task, as it mitigates the influence of class imbalance by treating each class equally. This is essential for cyberbullying detection, where the cyberbullying category often has significantly fewer samples than the non-cyberbullying category, making it critical to ensure that both classes are given equal importance during evaluation.

## Experimental results analysis

Table 4 presents the performance comparison of various models on the cyberbullying detection task, evaluated in terms of Macro-Accuracy, Macro-Precision, Macro-Recall,

**Table 4 Performance comparison of TANN and baseline models for cyberbullying detection, including Macro-Accuracy, Macro-Precision, Macro-Recall, and Macro-F1.** Bolded values indicate that TANN achieved the best performance among all models in terms of Macro-Accuracy and Macro-F1.

| Model | Accuracy | Precision | Recall | F1 |
|---|---|---|---|---|
| TextRNN | 0.8442 | 0.7746 | 0.7724 | 0.7735 |
| TextCNN | 0.8609 | 0.8089 | 0.7667 | 0.7844 |
| FastText | 0.8405 | 0.7751 | 0.7359 | 0.7520 |
| BERT | 0.8736 | 0.8134 | 0.8340 | 0.8229 |
| BERT-RNN | 0.8707 | 0.8098 | 0.8276 | 0.8181 |
| BERT-CNN | 0.8854 | 0.8482 | 0.8048 | 0.8234 |
| FLAN-T5 (Large) | 0.8736 | 0.8230 | 0.7960 | 0.8078 |
| BERT-FGM | 0.8853 | 0.8558 | 0.8116 | 0.8303 |
| TANN | **0.8895** | 0.8450 | 0.8278 | **0.8359** |

and Macro-F1. Overall, our proposed TANN model outperforms all baseline methods by achieving the highest Macro-Accuracy of 0.8895 and the highest Macro-F1 score of 0.8359.

The superior performance of TANN can be attributed to its hybrid architecture, which combines BERT with CNN and Bi-LSTM layers to capture both local and sequential features. Moreover, the use of a GRL as part of its adversarial learning strategy effectively reduces topic-specific biases, thereby enhancing the generalization capability of the model across diverse cyberbullying contexts.

Among the transformer-based baselines, two notable variants stand out. First, the BERT-CNN model, by merging the contextual embeddings of BERT with the ability of CNN to extract local features, achieves a strong Macro-Accuracy of 0.8854 and a competitive Macro-F1 of 0.8234. Second, BERT-FGM, which integrates adversarial training *via* the Fast Gradient Method, further improves robustness. FGM introduces small, calculated perturbations to the input embeddings during training, encouraging the model to learn smoother and more stable decision boundaries. Although its Macro-Accuracy of 0.8853 is comparable to that of BERT-CNN, BERT-FGM obtains a slightly higher Macro-Precision of 0.8558 and a Macro-F1 of 0.8303, highlighting the benefits of adversarial regularization in learning generalized features.

Another noteworthy baseline is FLAN-T5 (Large), an advanced large language model fine-tuned with instruction-based methods. Despite being applied in a zero-shot setting without additional domain-specific training, FLAN-T5 (Large) achieves a Macro-Accuracy of 0.8736 and a Macro-F1 score of 0.8078. These results are particularly impressive given that they rival those of fully supervised transformer-based models, underscoring the potential of instruction-tuned architectures for complex NLP tasks such as cyberbullying detection.

Traditional models such as TextRNN, FastText, and TextCNN consistently lag behind their transformer-based counterparts. Among these, TextCNN demonstrates the strongest performance, achieving a Macro-Accuracy of 0.8609 and a Macro-F1 score of 0.7844, likely due to its ability to efficiently capture local n-gram patterns. In contrast, FastText and

**Table 5 Statistical significance test results and confidence interval comparisons between TANN and baseline models.**

| Comparison | *t*-statistic | *p*-value | TANN 95% CI | Baseline 95% CI |
|---|---|---|---|---|
| *vs* TextRNN | 42.12 | $1.90 \times 10^{-6}$ | [0.8883–0.8907] | [0.8401–0.8490] |
| *vs* TextCNN | 60.44 | $4.49 \times 10^{-7}$ | [0.8883–0.8907] | [0.8593–0.8625] |
| *vs* FastText | 53.68 | $7.21 \times 10^{-7}$ | [0.8883–0.8907] | [0.8368–0.8442] |
| *vs* BERT | 24.12 | $1.75 \times 10^{-5}$ | [0.8883–0.8907] | [0.8706–0.8758] |
| *vs* BERT-RNN | 17.86 | $5.78 \times 10^{-5}$ | [0.8883–0.8907] | [0.8664–0.8751] |
| *vs* BERT-CNN | 11.66 | $3.00 \times 10^{-4}$ | [0.8883–0.8907] | [0.8838–0.8861] |
| *vs* FLAN-T5 (Large) | 38.98 | $2.59 \times 10^{-6}$ | [0.8883–0.8907] | [0.8723–0.8748] |
| *vs* BERT-FGM | 13.10 | $1.96 \times 10^{-4}$ | [0.8883–0.8907] | [0.8846–0.8859] |

TextRNN obtain Macro-F1 scores of 0.7520 and 0.7735 respectively, reflecting the limitations of earlier architectures in handling the nuanced and varied expressions characteristic of cyberbullying content.

In summary, the experimental results clearly indicate that transformer-based models, especially those enhanced with adversarial strategies such as in TANN and BERT-FGM, substantially outperform traditional approaches. TANN's superior performance, driven by its multi-level feature extraction and effective mitigation of topic-specific dependencies, validates its suitability for real-world cyberbullying detection in dynamic social media environments.

## Statistical significance tests

To rigorously assess the statistical significance of TANN's performance improvements, we conducted paired *t*-tests comparing TANN with eight baseline models: TextRNN, FastText, TextCNN, FLAN-T5 (Large), BERT-RNN, BERT, BERT-CNN, and BERT-FGM. As shown in Table 5, all comparisons yielded statistically significant differences at the $\alpha = 0.05$ level, with *p*-values spanning three orders of magnitude, ranging from $4.49 \times 10^{-7}$ for TextCNN to $3.00 \times 10^{-4}$ for BERT-CNN.

Confidence interval analyses further validate the consistent superiority of TANN. TANN exhibits a remarkably tight 95% confidence interval ranging from 0.8883 to 0.8907, with a range of only 0.0024, which does not overlap with the interval of any baseline model. In contrast, the classical models display substantially wider intervals: TextRNN ranges from 0.8401 to 0.8490 (range 0.0089), FastText from 0.8368 to 0.8442 (range 0.0074), and TextCNN from 0.8593 to 0.8625 (range 0.0032). Among the transformer-based models, FLAN-T5 Large ranges from 0.8723 to 0.8748 (range 0.0025), BERT-RNN from 0.8664 to 0.8751 (range 0.0087), and BERT from 0.8706 to 0.8758 (range 0.0052). Even the tighter intervals of BERT-CNN (0.8838 to 0.8861, range 0.0023) and BERT-FGM (0.8846 to 0.8859, range 0.0013) remain fully disjoint from that of TANN. Notably, BERT-FGM's upper bound of 0.8859 is still 0.0024 lower than TANN's lower bound of 0.8883.

The *t*-statistics further underscore the magnitude of these differences, ranging from 11.66 for BERT-CNN to 60.44 for TextCNN. Even BERT-FGM, the closest competitor

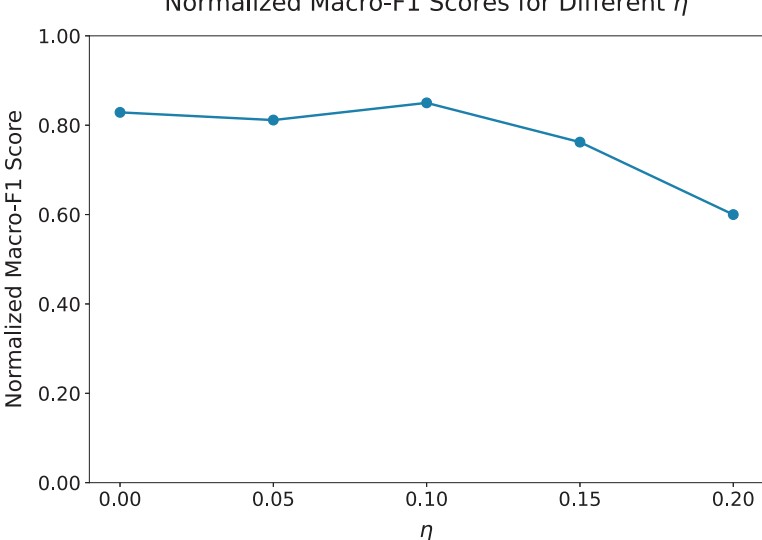

**Figure 3 Macro-F1 scores normalized to [0.6, 0.85] for different values of η.** The x-axis represents $\eta$ values ranging from 0.00 to 0.20, and the y-axis displays normalized Macro-F1 scores scaled to the [0.6, 0.85] range for clearer trend visualization. The optimal value is $\eta = 0.1$, which balances topic neutrality and effective cyberbullying detection. As $\eta$ increases beyond 0.1, performance declines, suggesting that stronger adversarial signals suppress critical features. When $\eta$ is too small, such as $\eta = 0.05$, the model overfits to topic-specific patterns.

with a *t*-statistic of 13.10 and a *p*-value of $1.96 \times 10^{-4}$, exhibits statistically inferior performance compared to TANN.

Collectively, these results provide robust evidence that the performance improvements of TANN are both statistically significant and practically meaningful. The consistent pattern of non-overlapping confidence intervals across all comparisons confirms the superiority and enhanced stability of TANN in cyberbullying detection tasks.

## Parameter sensitivity analysis

We first evaluated the impact of the hyperparameter $\eta$, which controls the strength of the adversarial signal in the GRL. We tested values of $\eta$ ranging from 0.0 to 0.2, as shown in Fig. 3. The results indicate that the model performs optimally when $\eta = 0.1$, achieving the best balance between enforcing topic neutrality and preserving key features for effective cyberbullying detection. As $\eta$ increases beyond 0.1, the performance of the model begins to decline, suggesting that overly strong adversarial signals can suppress features critical for accurate detection.

When $\eta$ is too small, as in the case of $\eta = 0.05$, the adversarial signal is weak, which allows the model to fit more topic-specific patterns, reducing its generalization across different topics. On the other hand, larger values of $\eta$, such as $\eta = 0.2$, overly penalize topic-related features, which diminishes the ability of the model to identify harmful content effectively. Thus, we chose $\eta = 0.1$ for all subsequent experiments, as it provides the optimal balance between generalization and robust detection.

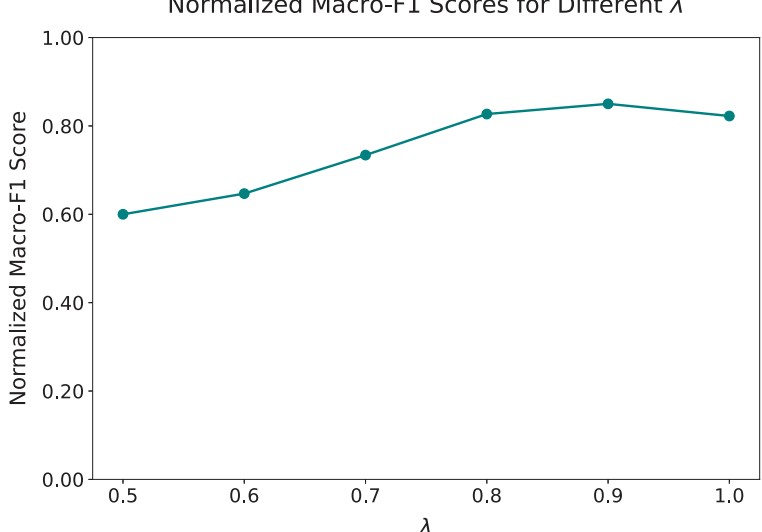

**Figure 4** **Macro-F1 scores normalized to the [0.6, 0.85] range for different values of $\lambda$.** The y-axis displays the full 0-1 scale while actual normalized scores occupy the [0.6, 0.85] subrange to accentuate performance variations. The graph illustrates the effect of varying $\lambda$ on the model's performance, with optimal balance between cyberbullying detection and topic generalization achieved at $\lambda = 0.9$. Performance improves with increasing $\lambda$ until peaking at 0.9, beyond which it stabilizes, indicating diminishing returns.               

Next, we examined the effect of the hyperparameter $\lambda$, which controls the trade-off between cyberbullying detection and topic-neutral feature learning. We varied $\lambda$ from 0.5 to 1.0, as shown in Fig. 4. The results show a steady improvement in the Macro-F1 score as $\lambda$ increases from 0.5 to 0.9, with the best performance achieved at $\lambda = 0.9$, where the score reached 0.8359. Beyond this point, increasing $\lambda$ to 1.0 leads to a slight decline in performance, suggesting that placing too much emphasis on detection reduces the ability of the model to generalize across topics.

The results highlight the importance of selecting an appropriate value for $\lambda$. Smaller values reduce the weight given to the detection task, limiting the ability of the model to accurately identify harmful content. Larger values overly prioritize detection, weakening the role of the adversarial mechanism in learning topic-independent features. Based on these findings, we set $\lambda = 0.9$ for all further experiments, as it offers the best balance between effective cyberbullying detection and cross-topic generalization.

Figure 5 shows the confusion matrix for the model with the optimal settings of $\eta = 0.1$ and $\lambda = 0.9$. The matrix indicates that the model correctly classifies 1,791 non-cyberbullying instances (Class 0) and 390 cyberbullying instances (Class 1), with relatively few misclassifications: 154 false negatives (cyberbullying misclassified as non-cyberbullying) and 117 false positives (non-cyberbullying misclassified as cyberbullying). This demonstrates the strong performance of the model, especially its ability to accurately identify harmful content.

These findings confirm the effectiveness of the adversarial learning strategy in reducing topic-specific interference while maintaining robust detection capabilities. The optimal

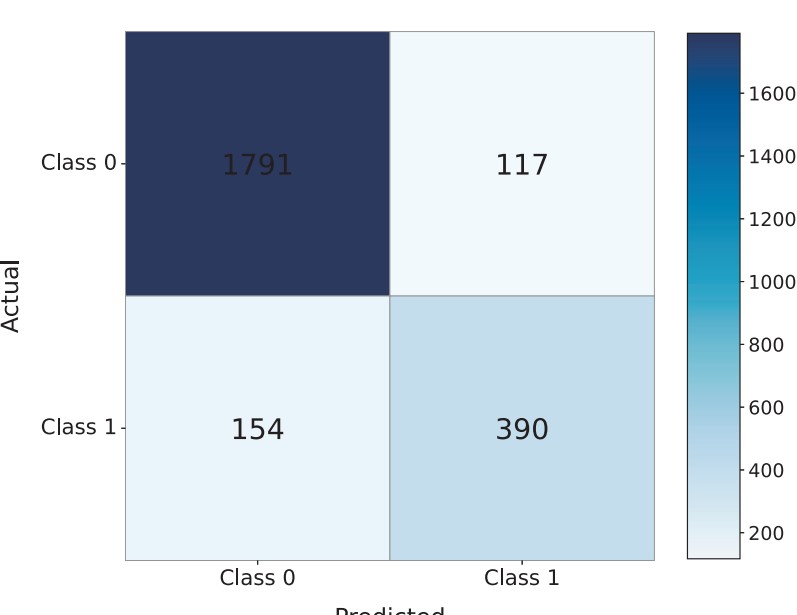

Confusion Matrix ($\eta = 0.1, \lambda = 0.9$)

**Figure 5 Confusion matrix for the TANN model with the optimal settings of $\eta = 0.1$ and $\lambda = 0.9$.** The matrix presents the classification performance of the model, showing the correct classification of non-cyberbullying (Class 0) and cyberbullying (Class 1) instances. Misclassifications include 154 false negatives and 117 false positives, highlighting the strong performance of the model in distinguishing between harmful and non-harmful content.

**Table 6 Performance comparison of full TANN and its variant without the topic adversarial mechanism.** The table reports precision and F1-scores for non-cyberbullying (Label 0) and cyberbullying (Label 1) detection, demonstrating that incorporating the topic adversarial mechanism enhances generalization and particularly improves detection for the minority cyberbullying class.

| Model | Precision (0) | F1-score (0) | Precision (1) | F1-score (1) |
|---|---|---|---|---|
| TANN (w/o topic) | 0.9188 | 0.9279 | 0.7628 | 0.7352 |
| TANN | 0.9208 | 0.9297 | 0.7692 | 0.7422 |

values of $\eta = 0.1$ and $\lambda = 0.9$ ensure a balanced approach, enabling the model to generalize effectively across diverse topics while accurately detecting cyberbullying. This balance is essential for real-world applications, especially on dynamic, multi-topic social media platforms.

## Ablation study

To understand how each module impacts performance, we conducted a series of ablation studies to evaluate the contribution of each component within the TANN model. We systematically removed key elements, including the topic adversarial mechanism, BERT embeddings, CNN layers, and Bi-LSTM layers. These experiments aimed to assess the impact of each component on the performance of the model.

**Table 7 Performance comparison of the full TANN model and its variants in the ablation study.**
Bolded values indicate that the full TANN model achieves the best performance across all variants in each metric.

| Model | Accuracy | Macro-precision | Macro-recall | Macro-F1 |
|---|---|---|---|---|
| TANN (w/o BERT) | 0.8768 | 0.8293 | 0.8013 | 0.8139 |
| TANN (w/o Bi-LSTM) | 0.8711 | 0.8095 | 0.8351 | 0.8210 |
| TANN (w/o CNN) | 0.8760 | 0.8186 | 0.8277 | 0.8230 |
| TANN (w/o topic) | 0.8866 | 0.8408 | 0.8233 | 0.8316 |
| TANN | **0.8895** | **0.8450** | **0.8278** | **0.8359** |

First, we removed the topic adversarial mechanism to create TANN (w/o topic). As shown in Table 6, the full TANN model outperformed this variant across all metrics. For non-cyberbullying detection, precision decreased from 0.9208 to 0.9188 and F1-score from 0.9297 to 0.9279. More significantly for cyberbullying detection, precision dropped from 0.7692 to 0.7628 and F1-score from 0.7422 to 0.7352. These results confirm the effectiveness of the adversarial mechanism in reducing topic-specific bias and improving generalization, particularly for the minority cyberbullying class with its diverse expressions.

Subsequent experiments removed other components as shown in Table 7. Removing BERT embeddings caused the most severe performance decline with a 2.3% accuracy drop, emphasizing their crucial role in capturing contextual semantics. Removing CNN layers reduced macro-precision by 3.2%, demonstrating their importance for local syntactic pattern extraction. Removing Bi-LSTM layers decreased macro-recall by 1.6%, highlighting their value in modeling sequential dependencies. The smallest performance drop occurred when removing the topic adversarial mechanism, with a 0.7% accuracy decrease, confirming its supplementary but valuable role.

These findings reveal three key insights. First, BERT embeddings form the foundation for semantic understanding. Second, CNN and Bi-LSTM layers provide complementary pattern recognition, with CNN focusing on local patterns and Bi-LSTM capturing sequential dependencies. Third, the adversarial mechanism enhances cross-topic generalization. The superior performance of the full model, with an accuracy of 0.8895 and a macro-F1 score of 0.8359, demonstrates the effectiveness of this component integration. Notably, while each element contributes individually, their combination achieves synergistic improvements, which is particularly critical for detecting nuanced cyberbullying patterns in real-world scenarios.

## Misclassification case analysis

To gain a deeper understanding of the model's limitations, this subsection conducts a qualitative analysis of the misclassification cases in the test set. Based on the types of misclassification errors, namely misclassifying samples with a true label of one as zero or misclassifying samples with a true label of zero as one, we summarize the following two typical error patterns and illustrate the potential shortcomings of the model with specific cases.

An analysis of the errors reveals that a primary challenge lies in misclassifications where the true label is 1 but the model incorrectly predicts 0. These false negatives often arise from semantic ambiguity and ironic expressions: some texts convey implicit negative emotions or sarcasm without using overtly negative terms, leading to misclassifications. For instance, the statement "这2个人还有完没完？赶紧该离离，该分分，不用告诉我们谢谢！没人在乎你们到底一起过不过" ("Are these two ever going to be done? Just hurry up and break up already—no need to announce it, thanks! No one cares whether you stay together or not.") may appear superficially neutral. However, the rhetorical questions imply a distinctly negative or sarcastic tone that the model fails to capture. Another crucial factor is the inability of the models to accurately detect negativity in domain-specific controversial events, primarily due to the lack of explicit sentiment words and insufficient contextual understanding, which leads to misclassification. For instance, the text refers to controversial behaviors such as the athlete's "false starts" and "false accusations," but because it does not directly employ negative sentiment words like "poor" or "shameful," the model fails to capture the negative connotations inherent in the context of the event.

Conversely, there are also instances in which the true label is 0 but is incorrectly predicted as 1, thereby exemplifying the issue of false positives. Such errors commonly arise from an overreliance on certain keywords or localized negative indicators, which can overshadow the broader neutral or positive context. For instance, the sample text "我的孙吧回来了" ("My 'Sunba' has returned") was misclassified as negative, potentially because the term "Sunba" had become associated with a negative subculture or community context. However, the text itself is purely neutral, containing no particular emotional stance. A comparable issue arises in the sentence "发个pyq怎么了，别人和母亲合个照你是不是觉得是显摆了？" ("What's wrong with posting on Moments? Just because someone takes a photo with their mother, do you think they're showing off?"), which contains a rhetorical question that could be mistakenly interpreted as negative. Nonetheless, the broader context merely addresses a neutral discussion of social media behavior. Despite the presence of rhetorical questions, the overall text provides an objective assessment of social conduct. However, the model may overemphasize emotionally charged, such as "showing off", and consequently misclassify the sentiment as negative.

These findings highlight the limitations of the model in comprehending context, especially with regard to irony, metaphor, or complex linguistic phenomena, and its inadequate adaptation to specific domains that require additional background knowledge. Data imbalance may further exacerbate these errors because samples containing subtle or implicit negative expressions tend to be underrepresented. Potential solutions include introducing more nuanced contextual encoding mechanisms such as enhanced attention, incorporating external knowledge modules for domain adaptation, and employing adversarial training or data augmentation to achieve more balanced distributions. This analysis underscores the need for refining fine-grained sentiment classification and improving domain generalization. Future endeavors may benefit from integrating multimodal information such as emoticons and event context to bolster robustness in complex classification scenarios.

## DISCUSSION

This study presented the TANN, a novel framework that integrates multi-level feature extraction using BERT, CNN, and Bi-LSTM, together with an adversarial learning strategy to mitigate topic-specific biases in cyberbullying detection. Experimental results demonstrate that TANN consistently surpasses both traditional and transformer-based baselines, particularly in terms of Macro-F1 score, which better reflects the effectiveness of the model under class imbalance conditions.

Ablation studies confirm that each architectural component plays a critical role. Removing the topic adversarial module resulted in a measurable decline in detection performance, especially for the minority cyberbullying class, thus affirming the importance of enforcing topic invariance. Likewise, excluding BERT embeddings caused the most substantial performance degradation, underscoring their value in capturing contextual semantics. The CNN and Bi-LSTM layers offered complementary strengths, namely local pattern recognition and sequential dependency modeling, respectively.

Hyperparameter sensitivity analysis further validated the robustness of the design. A moderate adversarial strength ($\eta = 0.1$) and a well-balanced trade-off between task performance and generalization ($\lambda = 0.9$) yielded optimal results. Excessively strong adversarial signals were found to suppress important discriminative features, while weaker signals failed to eliminate topic bias.

Error analysis revealed that most misclassifications stemmed from nuanced linguistic constructs, such as sarcasm, irony, or domain-specific expressions. False negatives typically involved subtle or implicit language, while false positives were often due to overemphasis on isolated negative keywords. These observations highlight limitations in contextual comprehension and domain adaptation. Future improvements could incorporate advanced attention mechanisms, external knowledge sources, or multimodal inputs (*e.g.*, emojis, metadata) to better handle such complexity.

Overall, the experimental findings and qualitative analyses confirm that the adversarial-cooperative training approach in TANN enhances both cross-topic generalization and model stability. These strengths make TANN a viable and robust solution for cyberbullying detection in real-world, multi-topic social media environments.

## CONCLUSIONS

In conclusion, this article presents TANN, an end-to-end adversarial framework designed to address the persistent challenge of cross-topic cyberbullying detection in social media. By integrating transformer-based embeddings with convolutional and recurrent layers in an adversarial training paradigm, TANN learns topic-invariant representations that generalize effectively across diverse topical domains.

Comprehensive experiments on a large-scale Chinese social media dataset demonstrate that TANN significantly outperforms both traditional and state-of-the-art models in terms of Macro-Accuracy and Macro-F1, highlighting its robustness and adaptability. The success of the model illustrates the efficacy of adversarial learning in mitigating

topic-induced bias, which is a critical factor in the generalization of text classification models across domains.

While TANN achieves strong overall performance, the misclassification analysis reveals remaining limitations in handling subtle and implicit linguistic cues such as sarcasm or rhetorical expressions. These limitations point to the need for enhanced contextual modeling.

To further improve performance and extend applicability, future work will investigate the integration of multimodal data sources (*e.g.*, emojis, user metadata) and advanced attention mechanisms. Additionally, extending the dataset to cover multiple languages and cultural backgrounds will be essential for testing the generalizability of the proposed method. Ultimately, TANN paves the way for building robust, generalizable cyberbullying detection systems capable of adapting to the dynamic nature of global online discourse.

### Funding
This work was supported by the MOE (Ministry of Education of China) Project of Humanities and Social Sciences (No. 24YJAZH149) and the Science and Technology Planning Project of Henan Province, China (No. 242301420143). There was no additional external funding received for this study. The funders had no role in study design, data collection and analysis, decision to publish, or preparation of the manuscript.

### Grant Disclosures
The following grant information was disclosed by the authors:
MOE (Ministry of Education of China) Project of Humanities and Social Sciences: 24YJAZH149.
Science and Technology Planning Project of Henan Province, China: 242301420143.

### Competing Interests
The authors declare that they have no competing interests.

### Author Contributions
- Shufeng Xiong conceived and designed the experiments, analyzed the data, authored or reviewed drafts of the article, and approved the final draft.
- Wenzhuo Liu performed the experiments, performed the computation work, authored or reviewed drafts of the article, and approved the final draft.
- Bingkun Wang conceived and designed the experiments, authored or reviewed drafts of the article, and approved the final draft.
- Yinchao Che performed the experiments, analyzed the data, performed the computation work, prepared figures and/or tables, and approved the final draft.
- Lei Shi conceived and designed the experiments, analyzed the data, prepared figures and/or tables, and approved the final draft.

## Data Availability

The data and code is available at Zenodo: henau-nlp. (2025). henau-nlp/TANN: v1 (Version v1). Zenodo. https://doi.org/10.5281/zenodo.15066726.

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
