# Peer review of "Topic adversarial neural network for cross-topic cyberbullying detection"

_PeerJ Computer Science, doi:10.7717/peerj-cs.2942_

## Round 0.1 · original submission · Major Revisions

· Academic Editor

Major Revisions

The reviewers liked what you are trying to say do in this work, however they find many deficiencies in how (well) you do it, or how you present things. Consider all the feedback carefully while preparing a revision.

Reviewer 1 ·

Basic reporting

Topic Adversarial Neural Network for Cross-Topic Cyberbullying Detection
Decision: Major Revision

The problem in question is relevant in the domain of adversarial Neural Network feature extraction towards the identification of cyberbullying over online traffic. Presenting such a study under the concept of a specialized solution of a feature extractor, cyberbullying detector and a topic discriminator is important for the continued development of adversarial feature extraction frameworks. The methodology developed and presented as the main contribution is described in a thorough and scientifically accepted manner as it is supported with a balanced equilibrium between theoretical and experimental presentation. My contribution through this review is concluded in a small number of remarks as follows:

1. In the abstract, the authors need to incorporate a concise although brief presentation of the paper’s contributions in the subject of “adversarial Neural Network feature extraction towards the identification of cyberbullying over online traffic”. This is a point that the authors need to focus and improve.

2. As it concerns the Related Work section this should be amended to include the most recent and prominent works on the field of neural network feature extraction over online traffic.

Also, a table in which the presented in the Related Work section published works would be briefly presented along with the chosen models and datasets per paper would be of great help for new readers.

3. Another important point that the authors should take under serious consideration is the possibility of any implications of errors within the selected datasets and more specifically regarding the chinese social media platform datasets. More information regarding the quality evaluation of a dataset could be found in this published work https://doi.org/10.1109/CNS56114.2022.9947235.

4. Moreover, a new table is needed to incorporate the metrices from the existed solutions and the one under review in order to allow the reader to compare the results.

5. Finally, on major concern is that no preprocessing of the Chinese social platform datasets is presented along with no details with the actual contents of each subset. Also, equally important is that the authors do not provide the source code of the proposed neural network solution nor the subsets upon which its’ evaluation was conducted. Overall, a good job that needs to get amended in order to be adequate for publication.

My finaly proposition is Major Revision.

Experimental design

The authors should refer to the "Basic Reporting"

Validity of the findings

The authors should refer to the "Basic Reporting"

Additional comments

The authors should refer to the "Basic Reporting"

Cite this review as

·

Basic reporting

- The paper is written in professional and clear English, but some minor grammatical errors and awkward sentence structures should be improved for readability.
- Some technical terms should be better explained for clarity.
- There are some problems with the citation where some citations are double. Maybe there are issues on the latex file. Please check.
- The paper provides an extensive literature review, covering traditional machine learning and deep learning models in cyberbullying detection.
- However, the discussion on domain adaptation and adversarial training could be better linked to cyberbullying detection.
- Most of the references on the literature review are already (quite) old (more than 5 years ago). Consider to add or update newer literature which closely related to this paper topics.
- The article follows a well-organized structure that conforms to PeerJ’s standards.
- Figures are relevant and well-labeled, but their descriptions could be more detailed in some cases. I also notice that some figures were not proportional (images are too big).
- The model architecture is well explained, but certain sections, such as the role of adversarial learning and its effect on cyberbullying classification, could be better elaborated. More explanation on how the topic adversarial neural network generalizes across topics would be helpful.

Experimental design

- The study is well within the journal’s scope and contributes to improving cyberbullying detection across different topics.
- The dataset description is very minimal. Authors only mention its source from Chinese social media platforms (Weibo and Tieba). However there is no explanation about the keywords
- The authors describe manual annotation, but they do not provide details on inter-annotator agreement, quality control, and annotators detail. This should be included to ensure labeling reliability.
- The methodology is generally well described, covering: 1. The Topic Adversarial Neural Network (TANN) architecture; 2. The three main components: feature extractor, cyberbullying detector, and topic discriminator; 3. The role of the Gradient Reversal Layer (GRL) in enforcing topic neutrality.
- However, the experimental setup is missing certain details, such as: 1. The hardware used for training (e.g., GPU specifications); 2. The training duration and computational cost; 3. The explanation of hyperparameter tuning beyond just listing tested values.
- The model configuration and experimental setup should be further explained to improve the reproducibility, which is very important.
- The study compares TANN against multiple baselines, including FastText, TextCNN, BERT, BERT-CNN, and BERT-RNN.
- However, there is no explanation on why these particular baselines were chosen over other adversarial learning approaches. The study would benefit from a comparison with domain adaptation techniques to justify the novelty of the approach.
- It will be also beneficial to use several LLMs technology in zero-shot settings as with this scenario would be also possible to predict both topics and cyberbullying context.

Validity of the findings

- The results indicate that TANN outperforms all baselines, achieving an accuracy of 0.8895 and a macro-F1 score of 0.8359.
- The results are statistically sound, but statistical significance tests (e.g., t-tests or confidence intervals) should be included to ensure the improvements are meaningful. Therefore, authors are recommended to add the statistical test.
- The ablation study demonstrates the importance of adversarial learning by comparing TANN with and without the topic discriminator. This is an important analysis, but it could be further extended by testing: 1. The impact of removing individual feature extraction components (BERT, CNN, Bi-LSTM); 2. A sensitivity analysis on how adversarial strength (η) affects the model performance.
- The confusion matrix (Figure 5) is useful, showing the false positive and false negative cases.
- However, a qualitative error analysis (i.e., analyzing misclassified cases) would strengthen the discussion. The authors could provide examples of misclassified instances to highlight the limitations of the model.
- The authors acknowledge the limitations, such as the dataset’s focus on Chinese social media, which may limit generalization to other languages.

Additional comments

Overall, the paper presents an interesting direction for cross-topic cyberbullying detection, but in its current form, it is not yet ready for publication. Below I will list the area could be improved in this draft of paper for further submission:
- While the paper introduces an adversarial learning framework for cross-topic cyberbullying detection, similar ideas have been explored in prior works, particularly in multitask learning and adversarial domain adaptation approaches from 3-5 years ago.
- To strengthen the contribution, the authors should clearly highlight how this work advances beyond existing methods. A comparative discussion with previous multitask learning-based cyberbullying detection models would help clarify the novelty. Additionally, it would be beneficial if the authors provided more insights into what unique challenges their approach solves that prior methods did not address.
- The dataset building process is not sufficiently detailed. Information regarding how cyberbullying instances were identified, annotated, and labeled should be more explicitly described.
- The annotators' background, labeling guidelines, and potential challenges in annotation consistency should be discussed to ensure the reliability of the dataset.
- Additionally, without access to the dataset, replicating the study becomes difficult. If possible, making the dataset publicly available or providing a clear protocol for dataset reconstruction would greatly benefit future research.
- The experimental setup is not fully described, which raises concerns about replicability. Moreover, the source code has not been shared, making it difficult to verify the implementation and results. Providing open-source code on a platform like GitHub would be highly beneficial and improve the transparency of the work.

Cite this review as

---

## Round 0.2 · Major Revisions

· Academic Editor

Major Revisions

Thank you for the revision. The reviewer agrees that there is substantial improvement in the manuscript, yet, there remains some critical shortcomings. Specifically, it is unclear if there is any novelty in the work, and what if any real scientific challenge was addressed, and the contribution in doing so, with respect to the body of work that exists in this space. As such, the authors would need to revise the manuscript with an improved related works discussion which explains the problem and solution space, along with positioning of the current work in this space showcasing the gaps and challenges it addresses.

·

Basic reporting

1. The article is written in professional and clear English. Several grammatical and structural improvements have been made throughout the paper, although minor polishing may still be helpful.
2. The literature review has been significantly expanded, with the inclusion of more recent studies (2020–2024) in cyberbullying detection, adversarial learning, and domain adaptation. The authors also added a comprehensive summary table of related works and datasets.
3. The overall structure is clear and well-organized, aligning with PeerJ’s standards.
4. Figures have been resized for better proportionality. Figure captions and descriptions have been improved.
5. The definition of technical components (e.g., GRL, adversarial training mechanisms) is more clearly stated now. However, the paper does not include formal mathematical definitions or theoretical proofs, which would be expected if the authors were claiming methodological novelty.

Experimental design

1. The research problem is relevant and clearly defined, and the authors situate their work within the challenge of cross-topic generalization in cyberbullying detection.
2. However, the core model architecture (TANN) remains a combination of well-known components (BERT + CNN + Bi-LSTM with adversarial training). While effective, the novelty is incremental rather than conceptual.
3. The dataset creation process is now well explained, including:
- Cyberbullying keyword selection strategies,
- Annotator profiles and labeling guidelines,
- Inter-annotator agreement (Cohen’s kappa = 0.85),
- Annotation quality control.
4. The experimental setup includes:
- Hardware details (RTX 4060, 16GB RAM),
- Hyperparameter tuning via grid search,
- Training time and computational cost.
5. The dataset and source code are now publicly available via GitHub, addressing earlier reproducibility concerns.

Validity of the findings

1. The authors now include statistical significance testing (t-tests, p-values) to support their claims, addressing a critical gap from the previous version.
2. A more comprehensive ablation study is included, evaluating the contribution of each component (BERT, CNN, Bi-LSTM), and confirming the importance of adversarial learning.
3. The paper also presents a qualitative error analysis, including examples of misclassified instances, helping to contextualize the model’s limitations.
4. The authors benchmark their model against strong baselines, including traditional deep learning, adversarial learning (BERT-FGM), and large language models (FLAN-T5, zero-shot).
5. While the experimental results are robust, the novelty of the contribution still appears incremental, building on adversarial/multi-task domain adaptation frameworks proposed in prior works (3–5 years ago).

Additional comments

Authors have made commendable efforts in improving the reproducibility, evaluation rigor, and dataset transparency, but the novelty of the proposed TANN model remains limited. The architecture and learning strategy closely follow existing adversarial training and domain adaptation paradigms in cyberbullying or hate speech detection. However, the clarity of explanation, technical execution, and reproducibility have significantly improved. Therefore, I recommend rejecting the current version, but encouraging the authors to resubmit, either:
- With sharper positioning of the work as a practical engineering solution. OR
- By enhancing the methodological contributions in future iterations.

Cite this review as

---

## Round 0.3 · accepted · Accept

· Academic Editor

Accept

The authors appear to have addressed the critical concerns reasonably, and the manuscript now looks ready for acceptance.